# OpenReview forum: "Representation Geometry Mediates Neural Circuit Formation: Evidence from Systematic Regularization Analysis"
_ICLR.cc/2026/Workshop/Sci4DL — Sci4DL 2026_

### Official Review · Reviewer_HhWi · 2026-02-24

**Fit:** 3
**Significance:** 2
**Confidence:** 3

**Summary:**

This paper explores the connection between grokking and matrix condition number. It does this on two tasks, though mainly on the simpler binary arithmetic task.

**Strengths:**

The experimentation is rigorous and there is good evidence that the regularization technique proposed is effective. They claim to show causality but this reviewer could not understand the experimental procedures sufficiently to corroborate this.

**Suggestions:**

There is a lot in this paper and I did not properly follow their  mitigation experiments. What I thought the authors meant is that they explicitly control the condition number - how else would you explore causality? But I couldn't see how they did this, nor even if it is feasible. These experiments are also characterized by a lot a procedures (Cox, Barron-Kenny, mediation pathways. Sections 3.3 and 3.4 are really terse and - for this reviewer - too hard to reliably disambiguate.

It is challenging to write a 4 page paper. It should only tell one story. If you try to do more than that, each narrative thread is compromised.

The paper might also benefit from a clearer reflection on the value of the LinearSRIP technique: as presented it does not seem to offer any advantage over SRIP(spectral). Tied to this would be more insight into the performance importance of condition number being small -  how does it really matter if it is 2 vs 40 for example?

Last question: when the authors write about speedup, are they measuring only in terms of epochs (as seems to be the case) or is it actual amount of computation?

---

### Meta-Review · Area_Chair_VZmw · 2026-02-28

**Recommendation:** Accept

**Metareview:**

This only got one review. I find this review plausible, and taking it at face value, the paper's indeed studying the science of DL and is above the bar for acceptance.

---

### Decision · Program_Chairs · 2026-03-02

Accept